# Impact-Resistant and Tough 3D Helicoidally Architected Polymer Composites Enabling Next-Generation Lightweight Silicon Photovoltaics Module Design and Technology

**DOI:** 10.3390/polym13193315

**Published:** 2021-09-28

**Authors:** Arief Suriadi Budiman, Rahul Sahay, Komal Agarwal, Gregoria Illya, Ryo Geoffrey Widjaja, Avinash Baji, Nagarajan Raghavan

**Affiliations:** 1Industrial Engineering Department, BINUS Graduate Program-Master of Industrial Engineering, Bina Nusantara University, Jakarta 11480, Indonesia; ryo.widjaja@binus.ac.id; 2Xtreme Materials Lab, Engineering Product Development, Singapore University of Technology and Design (SUTD), Singapore 487372, Singapore; komal_agarwal@alumni.sutd.edu.sg; 3Physics Department, Matana University, Tangerang 15810, Indonesia; gregoria.illya@matanauniversity.ac.id; 4Mechanical Engineering, La Trobe University, Melbourne, VIC 3086, Australia; A.Baji@latrobe.edu.au; 5Nano-Macro Reliability Lab, Engineering Product Development Pillar, Singapore University of Technology and Design (SUTD), Singapore 487372, Singapore; nagarajan@sutd.edu.sg

**Keywords:** 3D helicoidal architecture, fiber-based polymer composite, impact resistance, lightweight photovoltaics (PV), integrated PV rooftop

## Abstract

Lightweight photovoltaics (PV) modules are important for certain segments of the renewable energy markets—such as exhibition halls, factories, supermarkets, farms, etc. However, lightweight silicon-based PV modules have their own set of technical challenges or concerns. One of them, which is the subject of this paper, is the lack of impact resistance, especially against hailstorms in deep winter in countries with four seasons. Even if the front sheet can be made sufficiently strong and impact-resistant, the silicon cells inside remain fragile and very prone to impact loading. This leads to cracks that significantly degrade performance (output power) over time. A 3D helicoidally architected fiber-based polymer composite has recently been found to exhibit excellent impact resistance, inspired by the multi-hierarchical internal structures of the mantis shrimp’s dactyl clubs. In previous work, our group demonstrated that via electrospinning-based additive manufacturing methodologies, weak polymer material constituents could be made to exhibit significantly improved toughness and impact properties. In this study, we demonstrate the use of 3D architected fiber-based polymer composites to protect the silicon solar cells by absorbing impact energy. The absorbed energy is equivalent to the energy that would impact the solar cells during hailstorms. We have shown that silicon cells placed under such 3D architected polymer layers break at substantially higher impact load/energy (compared to those placed under standard PV encapsulation polymer material). This could lead to the development of novel PV encapsulant materials for the next generation of lightweight PV modules and technology with excellent impact resistance.

## 1. Introduction

Among other renewable energy sources, PV can be considered as the most versatile—it can be used in highly urbanized areas, as well as in the most remote areas, and can even float in water (oceans, lakes, etc.). Therefore, it is critical to develop PV modules and technologies that are appropriate for the particular uses and their unique circumstances/conditions. A single PV module design and technology may not be appropriate for PV modules used in various applications. Lightweight photovoltaics (PV) modules, for instance, are important for certain segments of renewable energy markets. Many large- scale buildings—such as exhibition halls, factories, supermarkets, farms, etc.—have huge footprints, with a limited number of supporting pillars, hence the roof structure has low-load bearing capacity. Such roofs require lightweight PV modules, otherwise the expensive reinforcement of such building structures required before the installation of the heavy conventional glass-based PV modules would render the whole renewable energy project (building plus the PV power source) uneconomical and make it unattractive to potential business interests [1,2,3]. Lightweight PV as part of building-integrated PV and mostly for urban building applications has been discussed quite extensively and comprehensively elsewhere [3,4,5,6,7].

However, in the present manuscript, we propose another important role of lightweight PV that could contribute to the climate and sustainability challenge in the world through use in unique geography of countries like Indonesia. Lightweight PV modules would also be critical for accelerating the adoption of renewable energy in certain geographies, such as archipelagic Indonesia—where many underdeveloped areas (which need renewable energy the most) are located in very remote locations and on thousands of separate islands. Heavy conventional PV modules would again render the prospects of building green renewable energy parks in such places uneconomical and unattractive due to very high transportation and installation costs of such PV systems.

When it comes to accelerating the adoption of renewable energy to meet the climate sustainability challenge facing the world, Indonesia is an interesting case in point. The country is blessed with diverse and abundant energy sources—both renewable (wind, hydro, photovoltaic, geothermal) and fossil. However, Indonesia’s geographic conditions are less than ideal for efficient energy distribution. Indonesia is an archipelagic country with large, sprawling geographic regions, typically lacking electrical infrastructure in very remote, underdeveloped, and outermost areas, which are often separated by seas [8,9,10,11]. Centralized energy sources are not the ideal option for such a geography; independent, decentralized power generation based on small wind turbines or micro-hydropower plants combined with photovoltaic (PV) plants are. Small villages in remote areas are currently either cut off from a centralized power supply or run diesel generators. Such independent renewable energy systems, particularly in these remote areas, have a strategic importance for Indonesia as a country, and perhaps more importantly as an integrated part of the global economic and environmental ecosystem for sustainability. This challenge represents an opportunity to collectively transform the energy sector in Indonesia into a sustainable and environmentally friendly energy economy.

Lightweight PV can play a crucial role in setting up such independent energy supply systems that are needed in remote, rural areas, such as Indonesia, by saving transportation and installation costs of solar PV systems. The PV power supply systems must be lightweight to facilitate transportation to very remote areas that often lack road and mobility infrastructures. Therefore, lightweight PV modules are needed—whether for applications in urban buildings in most advanced countries (especially in Europe) [3,4,5,6], or for easy installation in the most remote and underdeveloped areas in unique geographies such as archipelagic Indonesia— for accelerating the adoption of renewable energy for the world. 

Since silicon is likely to be the mainstream PV technology for quite some time [3,4], we need to enable lightweight silicon-based PV modules. One of the major technical problems in designing lightweight PV modules is impact resistance and structural strength, especially against hailstorms and strong winds in countries with four seasons, such as Germany (or Europe in general) and in North America [12,13]. Although, the concept of lightweight flexible PV is very appealing, nevertheless, it is still not a viable solution due to issues with low module stiffness, and structural reliability [3,4,5,12,13]. Many commercial lightweight PV solutions (including ones following IEC/UL standards) have a limited operational lifetime [4,5]. Nevertheless, recent studies with significant material development and clever design have enabled wonderful enhancements in impact resistance of many polymer substrates used as the frontsheet (instead of glass) in existing lightweight PV modules [13,14,15,16], although the silicon cells inside remain fragile and highly susceptible to particular impact loads.

Upon impact loading (such as from hailstorms against the frontsheet of the PV module), the (non-glass) frontsheet itself maybe strong enough and does not break, but the energy is passed directly to the underlying materials, i.e., first encapsulant (ethylene vinyl acetate/EVA is the most typical in PVs), then eventually to the fragile silicon cells. Receiving the impact energy from the frontsheet, the EVA would just comply (it has high compliance) and thus the energy was transmitted further down to the silicon cells. The fragile silicon material is especially prone to such point impact loads, and thus either cracks occur (nucleates) or extend further than their propagation points [5,13,14]. Consequently, electrical performance (i.e., power output) will either reduce gradually (degrade over time) or drastically—possibly leading to hotspots and potential safety issues (such as fires, etc.). The standard PV test for this hail resistance of a PV module is known as the IEC 61215/61646 clause 10.17.

Natural structural materials found in mantis shrimp, nacre, and shells were recently reported to exhibit superior mechanical and especially impact characteristics [17,18,19]. For instance, the 3D architecture with helicoidal geometry found in the dactyl club of the mantis shrimp can dissipate energy through quasi-plastic compressive reactions, forming a fracture toughening obstruction to the propagation of microcracks during repeated impacts [20,21,22,23]. Our group’s recent publications reported higher impact performance/resistance of such materials [24,25]. The layered geometry consisting of 3D helicoidally aligned fibers of such materials would efficiently absorb the impact energy and transfer very little energy to the fragile silicon solar cells. This would enable a novel lightweight PV module designs (based on polymer front and backsheets) with enhanced impact resistance and structural integrity/reliability (especially against cracks in the silicon cell). Our aim in the present study is to present the evidence for the basic feasibility of the proposed concept, i.e., using 3D-architected layered polymer structures consisting of helicoidally aligned fibers to provide protection of the silicon solar cells (in lightweight PV module designs) especially against the initiation/propagation of cracks due to impact loads (from hailstorms, for instance). Building on our previous research investigations on these novel materials [24,25,26,27], we extend our methodologies to enable this feasibility study for the application of lightweight PV technologies. In addition, the design of lightweight PV modules would allow PV integration with curved or surfaces with contours (such as for automobiles or boats) thus enabling more aesthetic design for the integration of PV into urban structures or buildings in cities.

## 2. Materials and Methods

### 2.1. Materials

Polyvinyl alcohol (PVA) having M_W_ = 98,000, polyvinylidene fluoride-co-hexafluoropropylene (PVDF-HFP) having M_W_ = 400,000, acetone and dimethylacetamide (DMAc) were purchased from Merck, Singapore. These chemicals were then used without further purification. PVDF-HFP was then dissolved in 1:3 solvent ratio (wt/wt) of acetone and dimethylacetamide, which was then stirred overnight (temperature = 45 °C) to prepare 35 wt% PVDF-HFP (DMAc/acetone) solution be used later for electrospinning. The rheological properties of the PVDF-HFP (DMAc/acetone) solution allowed morphology change from round to broad and flatter fibers.

#### 2.1.1. Choice of Fiber Material

PVDF-HFP polymer possesses properties, such as higher solubility [28], greater free volume [29], better mechanical properties [30], ease of processing, and flexibility [31].

#### 2.1.2. Choice of Matrix Material

Several polymers were considered for embedding the PVDF-HFP fibers in the matrix. However, polymers like polyurethane, polyurethane acrylate, epoxy, etc. are viscous polymers and form thicker films in comparison to the fibers. In some of the polymers, the polymer is dissolved in a solvent to form a liquid matrix, which in turn destroys the structural design of the fibers. While the thermoset matrix materials deal with temperature curing or UV curing which just increases the number of optimization processes and also end up dominating the mechanical properties of the composites. For the amount of fiber samples that the equipment allows to be made in the laboratory setting, it is very important to choose the matrix material very wisely such that its viscosity can be altered to form thin-films without affecting the structure of the fibers.

Therefore, after several trials with varieties of matrix materials, PVA was chosen because of its solubility in water (H_2_O), and its ability to form uniform films. Further, PVA also provides the capability to control its rheological properties to control the weight ratio between the fibers and matrix material in the fabricated composite. PVA is also more compatible with several other polymer materials due to its hydrophilic nature and is also a transparent polymer. Thin-films can be prepared by simple water evaporation with no requirements of any external factors [24]. The choices of these materials are simply as model materials with the aim to demonstrate the basic feasibility of the concept of enhanced impact resistance through 3D architected encapsulant enabled by electrospinning-based additive manufacturing methodologies.

### 2.2. Electrospinning-Based Additive Manufacturing

Near-field electrospinning (NFES, designed and assembled at SUTD, Singapore [24]) has been used as an additive manufacturing technique to fabricate helicoidally aligned fibrous layers (HA-FLs). In NFES, a high voltage is applied between a hemispherical polymer drop impregnated by a needle and the metallic collector plate [32,33]. NFES deposits one-dimensional fibers in a controlled manner at precise locations that allow obtaining of 3D structures by stacking the fibers layer by layer with angular offsets, as shown in Figure 1. A 5 mL syringe (make: Terumo Corporation, Tokyo, Japan) with a 25 G needle was filled with PVDF-HFP (DMAc/acetone) solution. The solution was then dispensed using a syringe pump (model no: EQ -500SP-H, make: Premier Solution Pte Ltd., Singapore). The flow rate of the PVDF-HFP (DMAc/acetone) solution was maintained as 1 mL/h. During NFES, voltage is applied to the needle whereas the aluminum collector plate was grounded. The collector plate is used for collecting fibers. The motion of the collector plate was controlled by placing it on programmable XY stage (PI High Precision XY stage) (model no: C-891, make: Physik Instrumente (PI) GmbH & Co. KG, Karlsruhe, Germany). The motion of the collection plate is then controlled to obtain layers of electrospun fibers. The layers of electrospun fibers were then stacked at different angular orientations to obtain HA-FLs. The details of the fabrication of HA-FLs with different angular orientations can be found in Komal et al. [24]. The speed of the motorized XY stage (as well as the collector) was fixed at 200 mm/s. The initial distance between the collector plate, and the needle was 7 mm, which was later decreased with a decrement of 0.5 mm with the collection of each fiber layer. The initial applied voltage during electrospinning was 2.4 kV for the deposition of the first four layers, which was then increased to 2.6 kV for the deposition of the next four layers.

The helicoidal structure was then achieved by stacking layers of electrospun fibers at different angular orientations on top of each other (see Figure 1). In the case of 45° HA-PVDFs with 45° angular offsets, the fiber layers were deposited at 0°, 45°, 90°, 135°, 180°, 225°, 270°, and 315°. In this case, the final HA-FLs consist of eight fused layers. The fabricated HA-FLs were then dried in a controlled environment inside an oven for 1 h at 50 °C to remove residual solvents. Three such 8-layered HA-FLs were then stacked onto each other to obtain a 24-layered helicoidally arranged HA-FLs.

### 2.3. Fabrication of Helicoidally-Aligned Synthetic Structural Composites (HA-SSCs)

The fabricated 24-layered HA-FLs were then embedded into PVA matrix solution to fabricate HA-SSCs. A 5 wt% PVA solution (doped with a surfactant) was sprayed onto the HA-FLs. The spraying of PVA solution was performed at a fixed flow rate along the length of the HA-FLs to obtain HA-SSCs. The surfactant (Triton X-100, Merck KGaA, Darmstadt, Germany) is added to the PVA solution to enhance the interfacial adhesion between HA-FLs and the PVA matrix. The samples were then dried at room temperature (with 75% humidity) for 72 h to fabricate opaque HA-SSCs, which, of course, is not yet appropriate for full integration into PV module design applications. An optically transparent material would be needed for real PV application with comparable transmission of sunlight (in terms of intensity and range of suitable wavelengths). However, as explained in the Introduction section as well as in the beginning of the Materials and Methods section, the focus of the present study is to demonstrate the basic feasibility of the concept of enhanced impact resistance through 3D architected encapsulant, not the full technological integration in PV module design. 

The thickness of the HA-SSCs samples ranged from 230 to 250 µm. For a more detailed description of the synthesis of the composites, see our earlier report [24]. Of the many types of HA-SSC samples, we reported there [24], we used only HA-SSCs15 and HA-SSCs45—with fiber alignment in rotational angle every 15° and 45°, respectively, from layer to layer—in the present study.

### 2.4. Impact Testing 

The impact tests were performed using the ball drop method, which was also employed by Chen et al. for their electrospun nylon (fiber pattern)-epoxy (matrix)-based composite [34] as well as own previous work [24,25]. The ball drop method typically finds the height at which the PV solar cell protected by the sample would break upon impact.

Customized setup used for determining the impact strength of the specimens is shown in Figure 2. A standard steel ball (diameter = 7.14 mm, weight = 1.4 g) was dropped under gravity onto the sample to measure the height required to facture solar cell. The height was increased in 5 cm increments to determine the final height required to fracture the solar cell placed beneath the samples. A digital oscilloscope (model no: DL1620 −200 MS/s 200 MHz make: Yokogawa Electric Corporation, Tokyo, Japan) with a resolution of 0.5 V/division and a charge meter (make: Kistler, model: 5015) with a sensor sensitivity of −4.060 pC/N were used to measure the impact force required to fracture solar cell. The samples were glued to solar cells at 2 diagonal edge points to restrict their motion during impact measurements. An unprotected bare solar cell was used as a control sample during impact measurements. The solar cells used are typical commercially available monocrystalline type without any interconnect (a new cell was used for each ball dropping experiment/test).

This employed impact test methodology is used due to the absence of a quantitative impact fracture mechanics methodology for small samples such as polymeric composites. Typically, standard impact fracture mechanics tests such as Charpy, ballistic, and Izod tests are suitable only for relatively large samples [35,36,37]. The current impact test methodology is used to determine the fracture height, and subsequently, the specific potential energy required to fracture solar cells. [24,25]. All the values of height and calculated specific potential energy at which the silicon solar cell breaks are compared to gain deeper insights about the impact properties of bare silicon solar cells (without any protection), silicon solar cells under EVA (ethylene vinyl acetate—typical encapsulant materials in PV technology), and silicon solar cells under HA-SSCs (15° and 45°). 

## 3. Results and Discussion

### 3.1. Physical Characteristics of the HA-SSCs

The impact properties of the electrospun HA-SSCs can be modulated by varying the offset angle of the stacked fibrous layers. Here, two samples (HA-SSC15 and HA-SSC45) were tested for their impact properties. In the cases of HA-SSC45 and HA-SSC15, after the deposition of the first fibrous layer, the next layer was deposited with its longitudinal axis rotated by angular offset of 45° and 15°, respectively. HA-FLs were then obtained by stacking eight layers of fibers at a fixed offset angle. The offset angle was 45° for HA-SSC45 resulting in fibrous layers located at 0°, 45°, 90°, 135°, 180°, 225°, 270°, and 315°, Similarly, an offset angle 15° for HA-SSC15 resulted in fibrous layers located at 0°, 15°, 30°, 45°, 60°, 75°, 900, and 105°. Three such 8-layered HA-FLs were then stacked onto each other to obtain a 24-layered HA-FLs, which were later embedded in a PVA solution to obtain HA-SSCs. 

SEM and optical microscope images (see Figure 3) show the microstructure of HA-SSCs produced using NFES. The SEM images show that the formation of broad ribbons, as compared to circular fibers, were found to have better contact area than circular fibers. The samples shown in Figure 3 belong to the same batch, which led to the improvement in the fibers’ production, fiber adhesion as described in detail in our previous report [24]. Figure 3(a1,b1) (optical images) clearly show the helicoidal arrangement of the ribbons in HA-SSCs. Figure 3(a2,b2) (SEM images) also show 24-layered HA-SSC. SEM images clearly show that ribbons were deposited at certain offset angles within the HA-SSC. The red dashed lines in Figure 3 show the angular sequence between each layer of the ribbons. For example, Figure 3(a1,a2) show that layers of the ribbons were oriented with 45° offset angles to obtain HA-SSC45. Similarly, Figure 3(b1,b2) show that layers of the ribbons were oriented with 15° offset angles to obtain HA-SSC15. 

### 3.2. Impact Test of Photovoltaic (PV) Cells

Silicon solar cells are fragile and are highly susceptible to impact load. Therefore, a customized impact testing setup (see Figure 2) was used to determine the impact resistance of such solar cells when protected by the samples [35]. A bare solar cell was chosen as a control group to compare with the solar cells protected with the fabricated samples. A standard steel ball (diameter = 7.14 mm, weight = 1.4 g) was dropped under gravity onto the specimens during impact test. At least six impact tests were performed for each type of sample (cross-sectional area = ~1.1 × 1.1 cm^2^). The height from which the ball was dropped was increased in increments of 5 cm to determine the fracture height of the solar cell placed under the sample. When the tentative fracture height for the samples was identified, the tests were then performed with 1 cm increments for careful estimation of the final fracture height. Table 1 summarizes the results of the impact tests of bare silicon solar cells (without any protection), silicon solar cells under EVA (ethylene vinyl acetate—typical encapsulant materials in PV technology), and silicon solar cells under HA-SSCs (15° and 45°) when impacted with the steel ball. The error bars in Table 1 show the range of the mean impacted height accumulated by performing experiments on at least six samples for each test condition. 

The fracture height means obtained experimentally are shown in Table 1. The nominal height of 25 cm in the bare Si solar cell group, for instance, was the actual height at which all silicon cells broke in the minimum six times we repeated the tests (often we conducted the ball drop tests on more than 6 samples at this height—up to 11 samples). When we did the test at a height of 20 cm (i.e., =25 − 5 cm), at least 50% of the silicon cells broke (out of the minimum 6 tests). At height of 30 cm (i.e., =25 + 5 cm), all silicon solar cells broke (out of the minimum 6 tests). Thus, the fracture data above merely shows the heights at which fractures began to be observed. Obviously, larger heights than the reported data above would break the silicon cells.

The statistical summary of the fracture height data of all the ball dropping experiments when fracture was observed in the underlying silicon solar cells are listed in Table 2. Each of the ball dropping tests was done with at least six samples and each sample consisted of a new silicon solar cell and protection layer each time. The means here were calculated considering the frequency at which fracture of the silicon cells happened at each height the experiment was conducted. The data here (in Table 2) shows an excellent agreement with the more discrete data shown in Table 1 (following the discrete heights as observed in the experiments).

Further, we conducted a statistical F-test with ANOVA (analysis of variance) method using level of significance = 0.05 to split observed aggregate variability found in the fracture height data into two parts: systematic factors and random factors (i.e., errors). The ANOVA test suggests that the treatments (i.e., the different protection layer placed on top of the silicon solar cell) are responsible for 98% of the variability found in the experiment, which we consider significant. This strongly suggests that the means of each group are significantly different from each other, and most of the variations found in the experiments were due to the different treatments in each group. In addition to this global ANOVA test, we also conducted individual tests comparing each group with each other group—one on one. The results suggest consistent findings with the global test. Hence, within the level of significance taken (i.e., 0.05) in this statistical analysis, each group was found to be significantly different from the others—even between HA-SSC15 vs. HA-SSC45 (which appeared to be the closest to each other amongst all other groups). This is certainly to be expected given the data as shown in Table 1 and Table 2.

### 3.3. Fracture of the Silicon Photovoltaic (PV) Cells

The optical microscope images after the impact testing of the representative silicon solar cells (front and back surfaces) are shown in Figure 4, Figure 5 and Figure 6. Each time the ball dropping experiment was performed, a new silicon solar cell and a new protection layer were used. All silicon monocrystalline solar cells used in the present study came from the same batch of 200 cells. Within the expected manufacturing variability specifications, all the silicon solar cells used in the present impact test/experiment may reasonably be assumed to have uniform mechanical strength, structural integrity, and fracture toughness. Thus, any differences we see after the impact experiments (as shown in Table 1, as well as in Figure 4, Figure 5 and Figure 6) may be associated with the protection/encapsulation samples (either EVA or HA-SSCs or nothing) placed on top of the silicon solar cells.

Both the results shown in Table 1 (the fracture height/specific potential energy data and the ensuing statistical analysis as summarized above) as well as the images shown in Figure 4, Figure 5 and Figure 6 clearly suggest that HA-SSC composite materials could provide better protection or encapsulation of the bare silicon solar cell against impact loads, compared to the standard PV EVA encapsulation layer. Both the HA-SSC composite materials (HA-SSC15 and HA-SSC45) allow substantially higher fracture heights and associated specific potential energies (significantly beyond the experimental error bars/uncertainties) before the silicon solar cells under them start breaking or initiating/propagating catastrophic fracture events upon the ball-dropping impact testing experiments.

When the EVA receives the impact energy from the ball drop test, it would simply comply due to its very low elastic modulus and thus the energy (and damage) is passed down to the silicon cells. The fragile silicon material is especially prone to such point impact loads, and thus cracks occur and extend through the thickness of the silicon solar cell almost instantaneously at relatively low height of 50 ± 4 cm. This is also evident from the images shown in Figure 5, where most of the impact energy was absorbed by creating only one main line (through the thickness of the cell) of a very long crack (almost across the whole area of the cell, which is often considered to be the more serious concern in terms of its effect on electrical power output degradation).

In contrast, upon receiving the impact energy, both the HA-SSC membranes owing to their helicoidally aligned fiber-reinforced layered structures would dissipate the impact energy very efficiently and deflect the crack laterally by following the helical path of the fiber directions instead of breaking instantaneously in a straight fracture line across the thickness of the membranes. In fact, the crack transitions between the matrix material and the ribbons at different angles in each layer—layer by layer. Every time the crack tip extends into the next layer, it finds the fiber aligned at different angle of orientation and thus gets deflected laterally in every layer, and it gradually loses its primary driving force/energy to extend through the thickness of the membrane. This avoids the catastrophic failure of the HA-SSC membrane, but also absorbs more energy to deflect the crack along different directions and planes in every layer, and due to regular encounters with laminate modulus changes, much less impact energy/damage is transmitted to the underlying silicon solar cell. These results are consistent with the load dissipation mechanism proposed in our previous reports [24,25] of this novel 3D helicoidally fiber-aligned composite materials, which is particularly effective and suggestive of more effective dissipating of energy such that only when the steel ball is dropped from the height of 69 ± 2 cm and 82 ± 4 cm for the HA-SSC15 and HA-SSC45, respectively, it would break the silicon solar cell underneath it.

The efficient mechanism of load dissipation and effective absorption of damage/energy are also evident from the images shown in Figure 6. Upon finally breaking the silicon solar cells underneath them at much higher heights, both the HA-SSC membrane materials exhibit several crack lines on the silicon cells indicating highly dissipated impact energy and transfer of load to the sideways (evident from the crack lines at different angles on the surfaces of the silicon cells) as received by the silicon cells. These cracks appear to follow the preferred crystallographic directions of <110> associated with the weakest crystallographic planes of {111} in typical silicon monocrystalline wafers/solar cells, as has been widely reported in the literature for both PV and other silicon-based energy devices [38,39,40,41,42,43,44,45].

These crack deflection mechanisms, which are evident from the impact tests in the present study as well as complete mechanical and microstructural characterization studies conducted in our previous reports [24,25], suggest a higher toughness and higher material’s resistance against crack propagation. Upon impact, the ribbons of the HA-SSC, which are built at different angles from one layer to the next, orient themselves along the direction of the applied force. During the deformation caused by the impact, the helicoidally arranged ribbons show an efficient transfer of the load to the adjacent ribbons by sliding and continuously pulling the ribbons until they break. This explains the higher energy absorbed by the HA-SSC protecting solar cells. The collective effect of the structural design and the variations in the moduli leads to a higher specific toughness of the HA-SSCs [24,25,34].

The HA-SSC45 samples seem to be the most effective in continuously deflecting crack or impact damage along different angular directions. It is evident from the fracture height data that fracture of the silicon solar cells occurs at 82 ± 4 cm—the highest amongst the materials tested in the present study. The crack repeatedly encounters differences in the modulus of the ribbons and the matrix material. In HA-SSC45 the crack continues along one ribbon, encounters a deviation in modulus due to the existence of the matrix material, continues to penetrate the matrix, then comes across another ribbon that is in a different direction and plane, and then deviates from its initial path to follow a different path. The crack propagates in different planes and directions, turning and twisting inside and outside the ribbon and matrix phases, further stretching the ribbon short of the final catastrophic events of specimen failure. It follows that due to the presence of helicoidal network of ribbons in the composites, this continuous deflection of the cracks requires a higher energy to break, thus transmitting a minimum of impact energy/damage to the underlying silicon solar cells.

However, our impact experiments suggest a slightly different picture for the HA-SSC15. Fracture of the silicon solar cells underneath these samples occurs at lower heights (69 ± 2 cm) and thus lower specific potential energy (indicating lower impact energy/damage absorption rate). In HA-SSC15, the ribbons are closer together than in HA-SSC45, which is due to lower offset angle. When a crack begins to form, its path is hindered when it encounters an alteration in direction more frequently compared to the composites with larger offset angle (in the case of HA-SSC45) due to the variation in the modulus through the thickness of the sample. When the angles between the layers are larger, the crack can propagate in a straight path in the matrix until the next ribbon prevents it. Because of the smaller offset angle, the crack encounters variations in phase modulus much more frequently, which should delay the fracture events even more and absorb even more energy. These results were shown in our previous report [24] when we placed a piece of glass under the materials. The results there [24] showed that the 15° helicoidally aligned samples could absorb much higher impact energy compared to the 45° samples. Since we followed the same full procedure of the ball-dropping experiment with [24,25], including the same size of the steel ball, we believe that this deviation (in the impact damage absorption capacity between the 15 vs. 45 HA-SSC samples) maybe associated with the silicon monocrystalline solar cells placed under the HA-SSC samples. Compared to glass slip, the silicon monocrystalline samples have crystallographic dependence of mechanical properties, including fracture preferred occurrences. However, the full correlation between the helicoidal orientation and the fracture of the monocrystalline silicon solar cells needs further investigation and is beyond the scope of the present study.

### 3.4. Enabling Next-Gen Lightweight Photovoltaic (PV) Module Technology

It is clear from the results presented so far, that the HA-SSC composites/membranes are very effective in absorbing and dissipating impact energy/damage and thus could protect the fragile silicon solar cells under them from point impact loads that silicon PV is very prone to. The present manuscript aims to provide evidence of the feasibility of using HA-SSC polymer films for PV encapsulation materials to protect the fragile silicon solar cells, especially in the design of lightweight PV modules that are particularly vulnerable to impact damage, such as hailstorms, as illustrated by the IEC 61215/61646 Clause 10.17. While many recent studies have shown that many light polymer-based materials [12,13,14,15,16] may be used to increase the overall structural integrity and mechanical strength (including fracture and impact resistance) of lightweight PV technology, the encapsulants used in those studies are invariably EVA—maybe with different thicknesses, or a slight variation, such as low cure EVA [46].

Obviously, EVA is the encapsulant of choice in the previous studies due to its cost implication and manufacturing readiness of the overall lightweight PV module. Our findings as reported in the present manuscript offer preliminary evidence, from the basic technological feasibility point of view, to use other kinds of novel polymer films with unique 3D architecture to enable stronger and higher resistant (including against fracture and impact loading) silicon-based lightweight PV module design. The manufacturing and economic implications are beyond the scope of the present manuscript. 

However, other technological issues need to be addressed to fully enable this novel concept. First, the HA-SSCs fabricated in the present study were not transparent. It is certainly a must to have transparent protective layers on top of the silicon cells in PV modules. It should be noted that these experiments were conducted to verify the basic feasibility of incorporating HA-SSC thin-films into PV modules. Now that our findings have confirmed the basic feasibility, using the same electrospinning-based additive manufacturing (AM) methodology, we could find other polymers which we can be fabricated into transparent forms, such as nylon [47] and poly(methyl methacrylate) (PMMA) (or PMMA-based composites) [48,49]. Moreover, the interfacial adhesion must be good not only with the frontsheet, but also with the silicon solar cell itself [46]. Lastly, the novel materials may need to be further developed to maintain their 3D architecture upon lamination process [46,50]. This could be the path forward for future studies to further develop these novel 3D-architected polymer composites/materials for enhancing the use of the silicon-based lightweight PV modules and technology.

## 4. Conclusions

Three-dimensional helicoidally architected fiber-based polymer composites synthesized in the present study using an electrospinning-based additive manufacturing (AM) methodology were shown to have excellent impact energy/damage absorption and dissipation rates. Such helicoidally aligned synthetic structural composites (HA-SSCs), when placed over monocrystalline silicon solar cells, may provide superior protection to silicon solar cells against impact loads, which is done via a ball-drop experiment in the present study. During the ball-drop experiment, both HA-SSC composite materials (HA-SSC15 and HA-SSC45) allow substantially higher fracture heights of 69 ± 2 cm and 82 ± 4 cm, respectively, in compassion to 25 ± 5 cm, and 50 ± 4 cm for unprotected Si cells and EVA protected Si cells, respectively. The present results provide preliminary evidence of a basic technological feasibility of using the novel materials for PV encapsulation to enable the design and technology of lightweight silicon-based PV modules. Full economic considerations of the implementations of this concept remain to be conducted in future investigations. Further, these results provide a promising framework for the development of strong, impact resistant, and resilient tunable polymeric composites suitable for technological applications ranging from aerospace to flexible electronic devices.

## Figures and Tables

**Figure 1 polymers-13-03315-f001:**
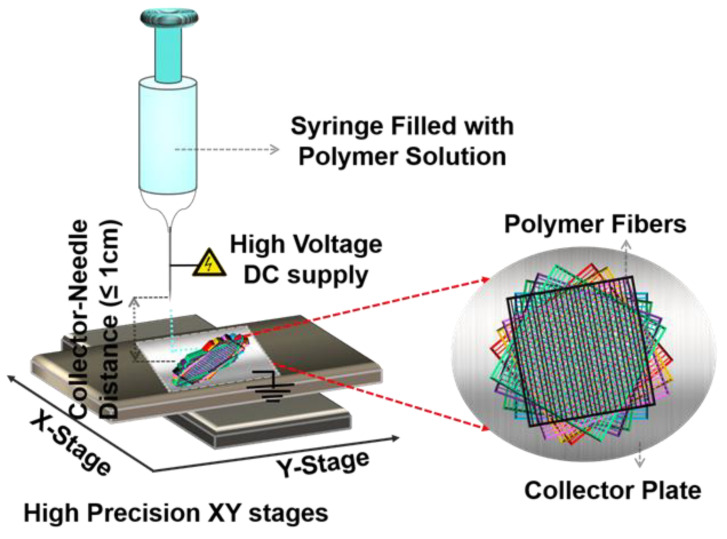
Schematic of NFES for depositing electrospun fibrous layers on a collector plate to fabricate 3D electrospun helicoidally aligned fibrous layers (HA-FLs).

**Figure 2 polymers-13-03315-f002:**
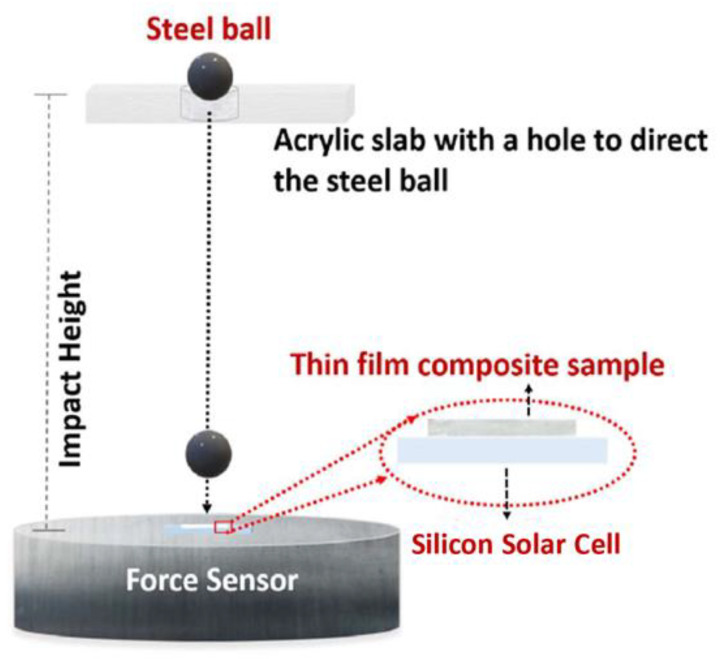
Schematic showing the impact test setup.

**Figure 3 polymers-13-03315-f003:**
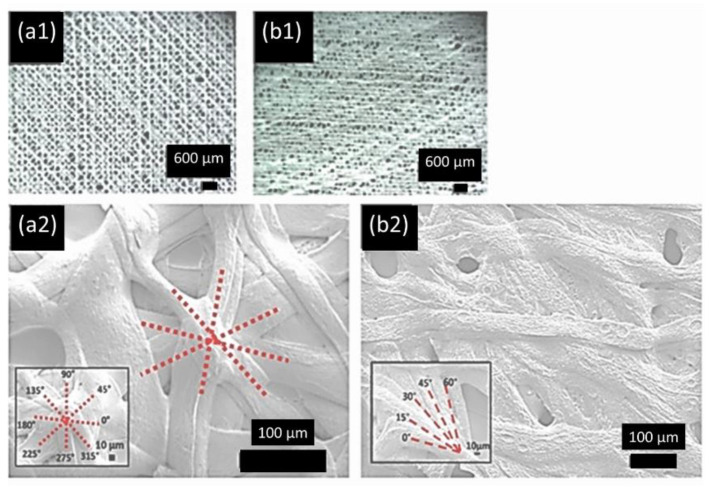
Optical micrographs at 35× of HA-SSC fabricated with offset angles: (**a1**) 45° and (**b1**) 15°. SEM images of HA-SSC fabricated with offset angle: (**a2**) 45° (at 270×) and (**b2**) 15° (at 190×). Reproduced with permission from American Chemical Society (ACS) [24].

**Figure 4 polymers-13-03315-f004:**
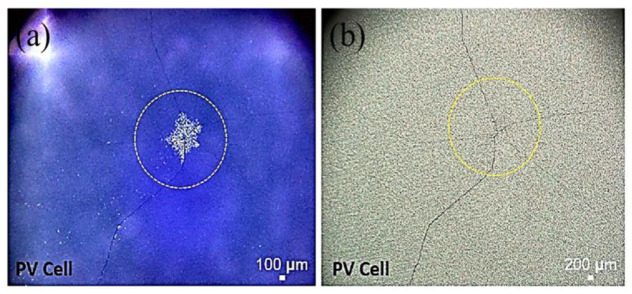
Microscope images taken after impact test: (**a**) front and (**b**) back of silicon monocrystalline solar cell after impact with a steel ball. These images were taken from a representative silicon solar cell broken at the nominal height as shown in Table 1 (in this case, this particular cell was broken when the steel ball dropped from a height of 25 cm). The circles on the silicon solar cell surfaces indicate the approximate impact contact area with the steel ball during impact test.

**Figure 5 polymers-13-03315-f005:**
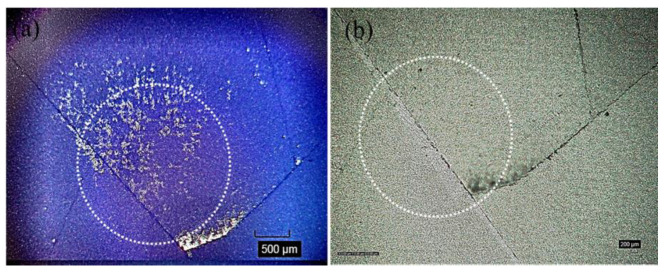
Microscope images taken after impact test: (**a**) front and (**b**) back of silicon monocrystalline solar cell after impact with a steel ball. The silicon cell was covered with EVA on top for protection against impacting ball. These images were taken from a representative silicon solar cell broken at the nominal height as shown in Table 1 (in this case, this particular cell was broken when the steel ball dropped from a height of 50 cm). The circles on the silicon solar cell surfaces indicate the approximate impact contact area with the steel ball during impact test.

**Figure 6 polymers-13-03315-f006:**
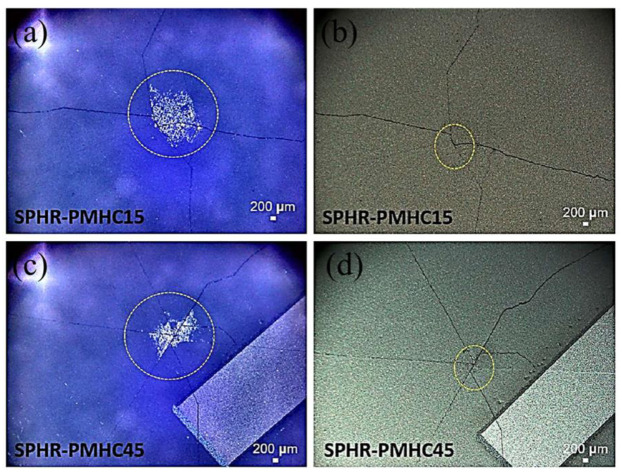
Microscope images taken after impact test: (**a**,**c**) front and (**b**,**d**) back of silicon cells after impact with the steel ball. The silicon cell was covered with HA-SSC on top for protection against impacting ball. For (**a**,**b**) the HA-SSC used was HA-SSC15 and (**c**,**d**) the HA-SSC used was HA-SSC45. These images were taken from the representative silicon solar cell broken at the nominal height as shown in Table 1. In this case, the particular cell in (**a**,**b**) was broken when the steel ball was dropped from a height of 69 cm, while the particular cell in (**c**,**d**) was broken when the steel ball was dropped from a height of 82 cm. The circles on the silicon solar cell surfaces indicate the approximate impact contact area with the steel ball during the impact test.

**Table 1 polymers-13-03315-t001:** Impact resistance properties of the samples protecting silicon solar cell.

Samples	Height at Which the Silicon Cell Breaks (cm)	Specific Gravitational Potential Energy (10^–2^ Jcm^3^/g)
Silicon solar cell	25 ± 5	-
EVA on Si cell	50 ± 4	7.3 ± 0.5
HA-SSC15 on Si cell	69 ± 2	9.4 ± 0.2
HA-SSC45 on Si cell	82 ± 4	11.2 ± 0.2

**Table 2 polymers-13-03315-t002:** Summary of the statistical analysis of the fracture height data.

	Bare Si Cell	Si Cell + EVA	Si + HA-SSC15	Si + HA-SSC45
Mean (cm)	25.39	48.97	70.05	82.06
Standard Deviation (cm)	3.19	2.71	2.19	2.29

## Data Availability

The data presented in this study are available upon request from the corresponding author.

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
