# Peer review of "Impact-Resistant and Tough 3D Helicoidally Architected Polymer Composites Enabling Next-Generation Lightweight Silicon Photovoltaics Module Design and Technology"

_polymers, 2021, doi:10.3390/polym13193315_

Round 1

Reviewer 1 Report

This manuscript describes the Impact-Resistant and Tough 3D Helicoidally Architected Polymer Composites Enabling Next-Generation Lightweight Silicon Photovoltaics Module Design & Technology. The work should be on the interest of the reader of the journal. There are, however, a number of points to be considered by the authors prior to possible publication of this paper. In short, the manuscript needs thorough revision in terms of language and clarity. The queries raised above also need to be addressed. Thereafter only the manuscript can be reviewed meaningfully. Hence the recommendation is to thoroughly revise the manuscript

  1. Abstract section is too long. Write very concise and specifically.
  2. Introduction section needs a major revision. This section is too long, need to rewritten in a standard way.
  3. Figure 3 caption is not the clear- check the figure.
  4. What is the specific advantage of this method compared recent reports on Si-solar cell.
  5. Lot of typographical errors and grammatical mistakes noticed in the manuscript. Need compete check for the entire manuscript.
  6. Conclusions should contain some quantitative information also

Author Response

Review report

Response to the reviewer’s comments on the Manuscript ID: polymers-1336104 titled “Impact-Resistant and Tough 3D Helicoidally Architected Polymer Composites Enabling Next-Generation Lightweight Silicon Photovoltaics Module Design & Technology,”

Reviewer 1

This manuscript describes the Impact-Resistant and Tough 3D Helicoidally Architected Polymer Composites Enabling Next-Generation Lightweight Silicon Photovoltaics Module Design & Technology. The work should be on the interest of the reader of the journal. There are, however, a number of points to be considered by the authors prior to possible publication of this paper. In short, the manuscript needs thorough revision in terms of language and clarity. The queries raised above also need to be addressed. Thereafter only the manuscript can be reviewed meaningfully. Hence the recommendation is to thoroughly revise the manuscript

Authors’ Response: The authors would like to thank the reviewer for his/her input towards helping us improve this manuscript. We appreciate very much the careful and comprehensive review provided, which have helped us tremendously in improving further the quality of the manuscript. Please kindly have below the list of our responses to his/her comments. The changes mentioned in this report and incorporated in the manuscript are highlighted by using blue font text.

Q1. Abstract section is too long. Write very concise and specifically.

Authors’ Response: The authors have carefully considered the reviewer’s comment and we have revisited and rewritten the abstract. It is now much shorter (249 words) and more concise and specific to the novelty and potential technological implications of our work, while keeping the essence and importance of the work as reported in the manuscript. Please have below the revised abstract as in the current revised manuscript.

Abstract, page 1, line 18: “Lightweight Photovoltaics (PV) modules are important for certain segments of the renewable energy markets – such as exhibition halls, factories, supermarkets, farms, etc. However, lightweight silicon-based PV modules have their own set of technical challenges or concerns. One of them, which is the subject of this paper is the lack of impact resistance, especially against hailstorms in deep winter in countries with four seasons. Even if the front sheet can be made sufficiently strong and impact-resistant, the silicon cells inside remain fragile and very prone to impact loading. This leads to cracks that significantly degrade performance (output power) over time. A 3D helicoidally architected fiber-based polymer composite has recently been found to exhibit excellent impact resistance, inspired by the multi-hierarchical internal structures of the mantis shrimp’s dactyl clubs. In that previous work, our group has demonstrated that via electrospinning-based additive manufacturing methodologies, weak polymer material constituents could be made to exhibit significantly improved toughness and impact properties. In this study, we demonstrate the use of 3D architected fiber-based polymer composite to protect the silicon solar cells by absorbing the impact energy. The absorbed energy is equivalent to the energy that would impact the solar cells during hailstorms. We have shown that silicon cells placed under such 3D architected polymer layers break at substantially higher impact load/energy (compared to those placed under the standard PV encapsulation polymer material). This could lead to development of novel PV encapsulant materials for next generation of lightweight PV modules and technology with excellent impact resistance.”

Q2. Introduction section needs a major revision. This section is too long, need to rewritten in a standard way.

Authors’ Response: Thanks again for the reviewer’s valuable comments and inputs. We have overhauled and rewritten many parts of the Introduction section. As the result, the Introduction section is now shorter and more systematic and logical as in the standard ways of Introduction. Please have below some parts of the Introduction section that we either shorten substantially or revised and improved significantly. These paragraphs are currently in blue fonts in the current revised manuscript. Of course, these are in addition to minor edits and revisions in many other places in the Introduction section as well as in the overall manuscript that are not necessarily highlighted in blue fonts.

Introduction section, Paragraph 3, page 2, line 66: “When it comes to accelerating the adoption of renewable energy to meet the climate sustainability challenge facing the world, Indonesia is an interesting case in point. The country is blessed with diverse and abundant energy sources – both renewable (wind, hydro, photovoltaic, geothermal) and fossil. However, Indonesia’s geographic conditions are less than ideal for efficient energy distribution . Indonesia is an archipelagic country with large, sprawling geographic– typically lacking electrical infrastructure in very remote, underdeveloped, and outermost areas which are often separated by seas [8-11]. Centralized energy sources are not the ideal option for such a geography; independent, decentralized power generation based on small wind turbines or micro-hydropower plants combined with photovoltaic (PV) plants are. Small villages in remote areas are currently either cut off from a centralized power supply or run diesel generators. Such independent renewable energy systems, particularly in these remote areas, have a strategic importance for Indonesia as a country, and perhaps more importantly as an integrated part of the global economic and environmental ecosystem for sustainability. This challenge represents an opportunity to collectively transform the energy sector in Indonesia into a sustainable and environmentally friendly energy economy.

Introduction section, Paragraph 5, page 2, line 91: “Since silicon is likely to be the mainstream PV technology for quite some time [3,4], we need to enable lightweight silicon-based PV modules. One of the major technical problems in designing lightweight PV modules is impact resistance and structural strength, especially against hailstorms and strong winds in countries with four seasons, like in Germany (or Europe in general) and in North America [12,13]. The idea of lightweight PV solution is very attractive but still not a viable option in the market due to problems with structural reliability and low module stiffness [3-5,12,13]. Many commercial lightweight solutions available in the market have a limited lifetime, even if the manufacturer claims compliance with IEC/UL standards [4,5]. Nevertheless, recent studies with significant material development and clever design have enabled wonderful enhancements in impact resistance of many polymer substrates used as the frontsheet (instead of glass) in existing lightweight PV modules [13-16], although the silicon cells inside remain fragile and highly susceptible to particular impact loads.”

Introduction section, Paragraph 7, page 3, line 116: “Natural structural materials found in bones, nacre, teeth, shells and mantis shrimp’s dactyl club have recently been reported to exhibit superior mechanical and especially impact characteristics [17-19]. For instance, the 3D architecture with helicoidal geometry found in shells such as Mantis shrimp’s dactyl club can dissipate energy from the impact region through quasi-plastic compressive responses, thereby providing a fracture toughening barrier against catastrophic propagation of micro-cracks when subjected to repeated impacts [20-23]. Our own group’s recent publications reported higher impact performance/resistance of such materials [24,25]. The layered geometry consisting of 3D helicoidally aligned fibers of such materials would efficiently absorb the impact energy and transfer very little energy to the fragile silicon solar cells. This would enable novel lightweight PV module design (based on polymer front and backsheets) with enhanced impact resistance and structural integrity/reliability (especially against cracks in the silicon cell ). Our aim in the present study is to present the evidence for the basic feasibility of the proposed concept – i.e., using 3D-architected layered polymer structures consisting of helicoidally aligned fibers to provide protection of the silicon solar cells (in lightweight PV module design) especially against the initiation/propagation of cracks due to impact loads (from hailstorms, for instance). Building on our previous research investigations on the novel materials [24-27], we extend our methodologies to enable this feasibility study for the application of lightweight PV technologies. In addition, the design of lightweight PV modules would allow PV integration with curved or surfaces with contours (such as for automobiles or boats) thus enabling more aesthetic design for the integration of PV into urban structures or buildings in cities.”

Q3. Figure 3 caption is not the clear- check the figure.

Authors’ Response: We have provided further information about the figure in the caption now and they are highlighted in blue fonts in the current revised manuscript. Figure 3 and the caption are now clear and visible as shown below

Figure 3. Low magnification top view optical microscope images (at 35X) of the 3D HA-SSC membranes oriented at: (a1) 45° and (b1) 15°. Higher magnification top view SEM images of the 3D HA-SSC membrane oriented at: (a2) 45° (at 270X) and (b1) 15° (at 190X). Reproduced with permission from the American Chemical Society (ACS) [24].

Q4. What is the specific advantage of this method compared recent reports on Si-solar cell.

Authors’ Response: The focus of the paper is to contribute further to the full feasibility of impact-resistant lightweight PV modules. Specifically, through the encapsulant material which we proposed in the present manuscript to be synthesized in 3D architecture via additive manufacturing methodologies. In addition, the impact resistance is currently still an issue even with state-of-the-art PV modules (with glass frontsheet). A novel encapsulant consisting of helicoidally architected fiber is being used to impart impact-resistant properties to the laminated solar cell. The specific advantage of using helicoidally architected fiber-based polymer composite as encapsulant has also been discussed in detail in the introduction section (paragraphs 4) and result and discussion (section 3.2) in the current revised manuscript.

Q5. Lot of typographical errors and grammatical mistakes noticed in the manuscript. Need compete check for the entire manuscript.

Authors’ Response: Thanks for the careful and comprehensive attention of the reviewer. The authors have carefully checked the overall manuscript again and the have thoroughly revised the manuscript for grammatical as well as typographical errors manually. In addition, we checked and corrected further by using Grammarly, an English correction software. Major parts of the manuscript are rewritten to further improve the essence and readability of the work presented in the manuscript.

Q6. Conclusions should contain some quantitative information also

Authors’ Response: The authors have carefully considered the reviewer comments. The conclusion has been modified to include quantitative information. The sentence below with key quantitative information has now been included in the Conclusion of the current revised manuscript.

Conclusion, page 14, line 548: During ball-drop experiment, both HA-SSC composite materials (HA-SSC15 and HA-SSC45) allows substantially higher fracture heights of 69 ± 2 and 82 ± 4, respectively in compassion to 25 ± 5, and 50± 4 for unprotected Si cell and EVA protected Si cell, respectively.”

Reviewer 2 Report

The manuscript entitled “Impact-Resistant and Tough 3D Helicoidally Architected Polymer Composites Enabling Next-Generation Lightweight Silicon Photovoltaics Module Design & Technology” by A. S. Budiman and co-workers is one of the extension of the author’s previous works (ACS Applied Polymer Materials 2020, Polymers 2020), however, this paper is enough of a newly developing extension of the previous efforts. And this work still contains some interesting elements for chemists working in this field. I believe this manuscript warrants the publication in Polymers after the following revisions.

  • Duplications of references 24 and 32 should be eliminated.
  • Reference 50 could not be found in the corresponding journal.

Author Response

Review report

Response to the reviewer’s comments on the Manuscript ID: polymers-1336104 titled “Impact-Resistant and Tough 3D Helicoidally Architected Polymer Composites Enabling Next-Generation Lightweight Silicon Photovoltaics Module Design & Technology,”

Reviewer 2

The manuscript entitled “Impact-Resistant and Tough 3D Helicoidally Architected Polymer Composites Enabling Next-Generation Lightweight Silicon Photovoltaics Module Design & Technology” by A. S. Budiman and co-workers is one of the extension of the author’s previous works (ACS Applied Polymer Materials 2020Polymers 2020), however, this paper is enough of a newly developing extension of the previous efforts. And this work still contains some interesting elements for chemists working in this field. I believe this manuscript warrants the publication in Polymers after the following revisions.

Authors’ Response: The authors would like to thank the reviewer for considering our manuscript for publication in the journal. The authors have carefully considered the reviewer comments and submitted herein a list of responses to the comments. The changes mentioned in this report and incorporated in the manuscript are highlighted in purple fonts.

Q1. Duplications of references 24 and 32 should be eliminated.

Authors’ Response: The authors thank the reviewer for the comment. The authors have made the necessary corrections in the manuscript and duplication of reference is removed from the manuscript.

Q2. Reference 50 could not be found in the corresponding journal.

Authors’ Response: The authors thank the reviewer for the comment. The authors have made the necessary corrections in the manuscript. The correct details of the reference are added in the manuscript and is as follows”

References, page 17, line 707: Wu, M.; Wu, Y.; Liu, Z.; Liu, H. Optically Transparent Poly(methyl methacrylate) Composite Films Reinforced with Electrospun Polyacrylonitrile Nanofibers, J. Composite Materials. 2012, 46(21), 2731-2738.

Reviewer 3 Report

In this manuscript, the authors reported the preparation of the PVDF fibrous layers via electrospinning. The resultant polymer composites were then used as impact-resistant materials to protect silicon photovoltaic. This study sounds interesting; however my major concerns are listed below, rising questions on the novelty and insufficient characterization. Specifically, my main criticism in this paper is that the reported composite layer is not transparent. Although the authors tried to explain the problem at the end of the article, they still couldn't convince me why would one make an opaque protector for PV? In addition, except for SEM, other basic spectrum characterizations of the polymer composites are missing.

Other issues:

The unit of Specific Gravitational Potential Energy looks wrong; it should be J/Kg or m2/s2.

The manuscript is not well written. For instance, line 333, “…as shown in Table 1 were as observed in the experiments The nominal….”

From my perspective, this study is not suitable for publication at this stage.

Author Response

Review report

Response to the reviewer’s comments on the Manuscript ID: polymers-1336104 titled “Impact-Resistant and Tough 3D Helicoidally Architected Polymer Composites Enabling Next-Generation Lightweight Silicon Photovoltaics Module Design & Technology,”

Reviewer 3

In this manuscript, the authors reported the preparation of the PVDF fibrous layers via electrospinning. The resultant polymer composites were then used as impact-resistant materials to protect silicon photovoltaic. This study sounds interesting; however my major concerns are listed below, rising questions on the novelty and insufficient characterization. Specifically, my main criticism in this paper is that the reported composite layer is not transparent. Although the authors tried to explain the problem at the end of the article, they still couldn't convince me why would one make an opaque protector for PV? In addition, except for SEM, other basic spectrum characterizations of the polymer composites are missing.

Authors’ Response: Thank you for the inputs and suggestions to further improve and explain about the opaque encapsulation material as reported in the present study. We appreciate it very much as it is important to clarify the focus and emphasis of this present work in the perspective of the overall development of the technology for the lightweight PV in the world of research and industry.

Our work is focused on the basic feasibility of the novel concept of an encapsulation material build and synthesized in 3D architecture via recently enabled electrospinning-based additive manufacturing methodology, as demonstrated in Sahay et al. 2020 and Agarwal et al. 2020 (references below), in enhancing the impact integrity of the lightweight PV module. This emphasis is on basic feasibility of the concept and full technological application of it is beyond the scope of this manuscript. It will need to be further studied and the materials, the processes as well as the integration to the overall lightweight PV module need to be further developed, as any new material introduction in the PV technology and industry.

The current material was chosen simply as the model material, as we have added now in the Materials section (Section 2.1.2, page 4, line 171) of the current revised manuscript as further explanation and clarification. Please see the additional sentence as currently in the revised manuscript as follows, shown in green color:

 “The choices of these materials are simply as model materials with the aim to demonstrate the basic feasibility of the concept of enhanced impact resistance through 3D architected encapsulant enabled by electrospinning-based additive manufacturing methodologies.”

The aim is to demonstrate the basic feasibility of the concept that the 3D helicoidally architected fiber-based polymer composite as encapsulant will provide better impact resistance in comparison to commonly used encapsulant such as EVA as composite encapsulant proposed here will have the capability to deflect and thereby hinder the crack propagation and thus will impart high impact resistance to the laminated PV modules. This has now been further emphasized with the additional sentences below in the 2.3 Fabrication of Helicoidally-Aligned Synthetic Structural Composites (HA-SSCs) (page 5, line 216) in the current revised manuscript.

This produced opaque PVDF-HFP ribbon reinforced PVA matrix synthetic structural composites (HA-SSCs), which of course is not yet appropriate for full integration into PV module design application. An optically transparent material would be needed for real PV application with comparable transmission of sunlight (in terms of intensity and range of suitable wavelengths). However, as explained in the Introduction section as well as in the beginning of the Materials section, the focus of the present study is to demonstrate the basic feasibility of the concept of enhanced impact resistance through 3D architected encapsulant, not the full technological integration in PV module design.”

Other polymer materials that are transparent have been fabricated using electrospinning techniques, as we have explained in the Section 3.3 of the manuscript. Electrospun polymer fibers such as Nylon [Ref 47 in the manuscript] and Poly(Methyl Methacrylate) (PMMA) (or PMMA-based composites) [Refs 48,49 in the manuscript] are transparent and would be suitable for PV application. The parts where we have discussed this in the manuscript may be found in Section 3.3 (page 13, line 525) and also included below:

However, other technological issues need to be addressed to fully enable this novel concept. First, the HA-SSCs fabricated in the present study were not transparent. It is certainly a must to have transparent protective layers on top of the silicon cells in PV modules. It should be noted that, these experiments were conducted to verify the basic feasibility of incorporating HA-SSC thin-films into PV modules. Now that our findings have confirmed the basic feasibility, using the same electrospinning-based Additive Manufacturing (AM) methodology, we could find other polymers which we can fabricate in transparent forms, such as Nylon [47] and Poly(Methyl Methacrylate) (PMMA) (or PMMA-based composites) [48,49]. Moreover, the interfacial adhesion must be good not only with the frontsheet, but also with the silicon solar cell itself [46]. Lastly, the novel materials may need to be further developed to maintain the 3D architecture upon lamination process [46,50]. This could be the path forward for future studies to further develop these novel 3D-architected polymer composites/materials for enhancing the use of the silicon-based lightweight PV modules and technology.

Thus, in principles, if we apply our electrospinning-based additive manufacturing methodology to these polymer materials, then we will have enhanced impact resistance with optically transparent material as an appropriate encapsulation material for PV application. Certainly, the electrospinning-based AM process parameters and instrumental design may need some engineering efforts such as adjustments and modifications, just like in any technological development stages in new material introduction as part of the research and development of new PV module design. Those full technological development is beyond the scope of this manuscript.

We fully understand the reviewer's concern that the non-transparent protective layer will affect the performance of the solar cell and have added more comprehensive explanations and clarifications throughout the manuscript about the scope of the present study. Nevertheless, we believe that the helicoidally architected fiber-based polymer composite as an encapsulant for PV module application is a significant milestone and deserves further investigations in providing enhanced impact resistance to the laminated PV modules. This novel encapsulant concept could provide the path forward for future studies to further enabling and enhancing the use of the silicon-based lightweight PV modules and technology.

Furthermore, our current work is an extension of our previous works (ACS Applied Polymer Materials 2020, Polymers 2020) [1, 2], In ACS Applied Polymer Materials 2020, we have conducted complete and comprehensive materials characterizations of the helicoidally 3D architected fiber-based polymer composite (chemically, physically as well as mechanically) in addition to the SEM/microstructural characterization that we included in the present manuscript. The focus of the current manuscript is to demonstrate the basic feasibility of the helicoidally 3D architected fiber-based polymer composite for the application of enhanced impact-resistance of silicon-based lightweight PV module design.

The authors have also carefully considered the other reviewer’s comments and submitted herein a list of responses to their remarks. The changes mentioned in this report and incorporated in the manuscript are highlighted by using green font texts.

Other issues:

Q1. The unit of Specific Gravitational Potential Energy looks wrong; it should be J/Kg or m2/s2.

Authors’ Response: The authors understand review’s concern, nevertheless, here, we have divided Gravitational Potential Energy with density rather than mass of the sample, therefore our unit is Jcm3/g rather than J/Kg or m2/s2. We have performed similar calculations in our previous work [1,2]. Similar approaches have also been adopted by Chen et al. (ACS Appl Mater Interfaces 2019), which is Ref [34] in the current revised manuscript.

Q2. The manuscript is not well written. For instance, line 333, “…as shown in Table 1 were as observed in the experiments The nominal….”

Authors’ Response: The authors have carefully considered the reviewer’s comment and the authors have thoroughly analysed the manuscript for grammatical as well as typographical errors manually as well as by using Grammarly, an English correction software. Major parts of the manuscript are rewritten to further improve the essence and readability of the work presented in the manuscript.

The abovementioned line is rewritten as The fracture height means obtained experimentally are shown in Table 1in Section 3.2 (page 8, line 330)

 Q3. From my perspective, this study is not suitable for publication at this stage.

Authors’ Response: We appreciate the reviewer’s inputs and suggestions very much as it is important to further enhance the scientific quality and technological merits of this present work in the perspective of the overall development of the technology for the lightweight PV in the world of research and industry. The authors have carefully considered all the reviewer's comments and made substantial changes throughout the manuscript to improve the manuscript. Apart from incorporating all the recommendations from the reviews, the authors have also taken care of typographical as well as grammatical errors in the latest modified manuscript. The authors believe that the manuscript in the current form is a significantly improved version.

References

  1. Sahay R, Agarwal K, Subramani A, et al (2020) Helicoidally arranged polyacrylonitrile fiber-reinforced strong and impact-resistant thin polyvinyl alcohol film enabled by electrospinning-based additive manufacturing. Polymers (Basel) 12:. https://doi.org/10.3390/polym12102376
  2. Agarwal K, Sahay R, Baji A, Budiman AS (2020) Impact-Resistant and Tough Helicoidally Aligned Ribbon Reinforced Composites with Tunable Mechanical Properties via Integrated Additive Manufacturing Methodologies. ACS Appl Polym Mater. https://doi.org/10.1021/acsapm.0c00518

Round 2

Reviewer 3 Report

The authors have addressed some of my questions. This paper can now be accpted.